# The Politicization of Women's Health and Wellbeing

Udi Sommer [1,*] and Aliza Forman-Rabinovici [2]

[1]   Chair, Center for the Study of the United States at Tel Aviv University, School of Political Science,
      Government and International Relations, Tel Aviv University, Tel Aviv 6139601, Israel
[2]   School of Political Science, Government and International Relations, Tel Aviv University,
      Tel Aviv 6139601, Israel; alizaforman@gmail.com
*    Correspondence: udi.sommer@gmail.com

**Abstract:** The framers and advocates of the United Nations Sustainable Development Goals face a unique challenge when it comes to the goals of Sustainable Development Goal (SDG) 3, good health and wellbeing, as it concerns women's health. The health of women, and in particular reproductive rights, have been politicized in the work of the UN. Forums of the UN have become a battleground between those who would frame reproductive rights as a morality policy versus those who frame them as a feminist policy. This problem is not new to the organization's work. Indeed, it has been a challenge to the UN's ability to promote women's health for years. This article explores how the framing of women's reproductive rights poses a unique challenge to implementing some of the goals of SDG3, and in particular targets 3.1, 3.7, and 3.8. It also offers strategies to surmount the challenge with an example of a different intergovernmental organization that managed to overcome this issue.

**Keywords:** SDG3; SDG5; abortion policy; reproductive rights; public health; morality policy; feminist policy; medical frame; Maputo Protocol; International Conference on Population and Development Programme of Action (PoA)

---

## 1. Introduction

The goal of the United Nations Sustainable Development Goal (SDG) 3 is to promote "*healthy lives and wellbeing for all at all ages*". Within the SDG, targets, 3.1, 3.7, and 3.8 address women's health and specifically reproductive health either within the target itself or within the target indicators. This policy field, however, presents unique challenges. Women's health, and reproductive rights specifically, are not just a health policy issue. Rather, they fall into two other theoretical categories of public policy: morality policy and feminist policy. The empirical overlap between these two theoretical categories has led to the fierce politicization of women's health issues and stands as a considerable barrier to achieving women's health goals. This article, therefore, sets out to explore this conflict and ask "what are the unique difficulties in addressing SDG3 goals on women's health?" We propose that the difficulty in addressing women's health is a result of the overlap between morality policy and feminist policy but that an alternative to the current UN strategy may be able to overcome this conflict.

Public health researchers have often taken the approach that public health policy will be the linear outcome of the generation of increasingly reliable public health data. Barriers to effective public health policy can be overcome when more accurate and trustworthy findings are presented. This approach, though, ignores the ideologies and political considerations that shape all policy, including health policy [1,2]. It assumes a dialogue between public health professionals and medical researchers that discounts much of the surrounding political environment [3,4]. This need to consider public health policies beyond the health professional context is especially relevant for public health issues of women's care and reproductive rights. Reproductive rights are often addressed using a discourse unrelated to

health. Some, seeing it as a morality policy, address it through a religious or traditional value-based lens. Others, seeing it as a gender equality policy, address it as a question of female autonomy and women's rights. The clash between the feminist policy view on the one hand and that of the morality policy on the other often divides political camps and functions as a barrier to policymaking. It might also contribute to disregarding aspects of intersectionality in questions of reproductive health.

This problem is not new in the work of the United Nations. UN forums have served as battlegrounds between religious and feminist interest groups, as they compete to impose their frame onto the UN agenda. In an effort to create consensus and achieve universal acceptance, UN bodies have circumvented this clash by applying ambiguous and indecisive language to protocols that address reproductive rights. That said, if reproductive health continues to be framed along the morality vs. feminist policy clash, it will severely compromise the global hope of achieving SDG3.

Can this problem be overcome? To answer this question, we'd propose looking to the African Union, as the only intergovernmental organization (IGO) to date to decisively address reproductive health questions and to frame a policy blueprint that addresses abortion. Based on lessons learned from the success of the African Union (AU), UN bodies might adopt two steps; first, they can fully commit to a medical frame when addressing reproductive rights. Second, they can replace a strategy that aims for universal consensus with one that is more open to conflict. While they might lose some support, UN bodies would provide significantly more support for new national policies and grassroots movements that need policy legitimation.

In order to explore the difficulties in addressing reproductive health policy in general and in UN work specifically, we start by defining the theoretical categories in which reproductive policy may fall: morality policy and feminist policy. We then trace how the conflict between those categories has played out in the history of UN work including in SDG3. To map a route for change, we then examine the AU Maputo Protocol and discuss the importance of alternative strategies for addressing reproductive health within UN forums.

## 2. Reproductive Health and the Clash between Morality Policy and Feminist Policy

In order to understand the unique politicization of women's reproductive health, we have to understand the theoretical public policy contexts in which it is commonly addressed. Reproductive rights are seen as both a morality policy and a feminist policy in the academic literature and in public debate. The overlap between the two theoretical categories sets policymakers up for a clash between ideologies and interest groups each supporting one of the approaches [5].

Morality policy is a theoretical category of public policy that meets three criteria. First, a morality policy is a policy that touches upon a central element of religious doctrine or that religious doctrine commonly and decisively addresses [6,7]. Second, this policy addresses a topic that relates to a values question [8,9]. Finally, the topic is perceived as being technically simple. Because of their technical simplicity, citizens feel comfortable engaging in public debate on morality issues [10–12]. Policies commonly considered morality issues include capital punishment, homosexual marriage, and reproductive health law.

Reproductive health policy meets all three criteria of a morality policy. First, most religions have a stance on when life begins and therefore when it is legitimate to prevent the production of new life through contraception and abortion [13]. Second, both religion and public values often address gender roles and what constitutes a family. Women engaging freely in sex could, according to many, jeopardize family security and sexual morality. Therefore, policy that enables women to engage in sex, while preventing pregnancy and avoiding sexually transmitted diseases, can chip away at the integrity of the family and religious conceptions of female sexuality [14]. Elements of reproductive health, such as contraception and abortion, also subvert the emphasis religious dogma often places on the role of women as mothers. Women, as the primary caretakers and educators, have a crucial role in transmitting religious doctrine to the next generation. As a result, the idea of motherhood as the apex of the female experience is often deeply embedded and protected in religion. Mothers are the moral guardians and

teachers of culture and norms. Therefore, law regarding the role of women, particularly as mothers and caregivers, can be especially sensitive to religious influence [15,16]. Reproductive health issues are also thought of as easily comprehensible and therefore comfortable to debate.

On the other hand, reproductive health policy is considered a feminist policy. This policy family includes policies that addresses at least three of five ideas [17]: (1) the improvement of women's rights, status, or situation; (2) the reduction of gender-based hierarchies; (3) a focus on or reconsideration of the public–private sphere distinction; (4) a focus on both men and women; (5) policy that can be readily connected with feminist groups or movements.

Women's reproductive health policy, [17], squarely falls within the definition of a feminist policy, and within this category is considered a *body politic* policy. It aims to improve women's status and situation through improved health services. It aims to reduce gender hierarchies by recognizing female bodily autonomy as well as enabling sexual freedom, expression, and security for women. Reproductive health policy moves women's sexuality from the private sphere into the public sphere through states taking responsibility for women's health. Finally, issues of sexual health, access to contraception, and abortion law are commonly associated with and addressed by feminist movements.

Reproductive health policy's status both as a morality policy and as a feminist policy sets it on a crash course in public debate. Some choose to see it as a feminist policy, framing reproductive rights as a matter of human rights, bodily autonomy rights, health rights, and a gender equality agenda. On the flip side, religious groups and traditional voters see it as a morality policy and as a threat to religious dogmas, traditional family structure and roles, and an assault on more traditional values. As we demonstrate in the following section, all of this also plays out at the UN level and has affected the UN's ability to promote comprehensive reproductive health policy for years.

Both approaches also often fail to account for questions of intersectionality, such as how motherhood, pregnancy, and abortion are perceived for and by different groups. For example, disabled women experience a double stigma, in which as women they are expected to raise children, but as people with disabilities they are often viewed as asexual or incapable of fulfilling traditional feminine roles [18]. The feminist movement has at times failed to account for the needs of women with disabilities or the rights of the disabled [19], while morality-based movements fail to recognize unique forms of reproductive healthcare they might require. Additionally, neither movement accounts for the difficulty of access to healthcare that many disabled women experience [20,21].

Reproductive health policy's status as a health, morality, and feminist policy is expressed in the different frames that interest groups and lobbyists use to promote their stance on the issue. The morality policy position is expressed through the "religious/natural family" frame [22]. This frame relies on religious understandings of conception and viability. It removes pregnancy from an individual rights frame and places it in the context of women's responsibilities as child bearers and mothers. Feminist policy advocates adopt the "women's rights" frame. This frame is based on world-society norms of human rights and liberal understandings of female autonomy. Finally, those approaching abortion through the public health perspective will tend to apply the "medical frame". This frame relies on a position that sees doctors and medical professionals as authorities on pregnancies and as those responsible for diagnosing and responding to medical complications. It is based on the public's faith in doctors' medical expertise and a respect for the doctor–patient relationship.

The choice of frame has been found to influence outcomes in campaigns for policy change. The medical frame is often the most successful in generating more permissive reproductive rights policy. Government officials can accept medical justifications for policy, while appearing rational and apolitical and avoiding controversy [22]. Nongovernmental organizations (NGOs) are more likely to get support from international organizations such as the UN or the European Union (EU) to increase pressure on reluctant governments to change policy when they come from a public health perspective [23].

### 3. The Clash between Morality and Feminist Policy Advocates in UN Work and the SDGs

This conflict over reproductive rights and how to address them has played out in UN work for over 30 years. While the UN has been a leader in determining the global gender equality agenda, it has had particular difficulty addressing women's reproductive health. Efforts to do so have become battlegrounds between pro-life and pro-choice interest groups, all vying to frame the policy agenda. The UN's method to negotiate the conflict has consistently been to resort to ambiguous language, which neither takes a decisive stand on reproductive health policy nor provides countries with clear guidelines on the topic. In this section, we trace this process through the 1994 International Conference on Population and Development Programme of Action, and up till the SDGs, to see how this had played out.

In 1994, the UN organized the International Conference on Population and Development (ICPD). Held in Cairo, it was attended by delegations from close to 200 countries and representation from around 1700 organizations [24,25]. The goal of the ICPD was to address development issues including human rights, population growth, sexual and reproductive health, gender equality, and sustainable development. It resulted in the writing of the ICPD Programme of Action (PoA), which was adopted unanimously by the UN member states in attendance and overall by 179 member states [25]. It is seen as the first time a major international agreement addressed women's rights and reproductive rights as an integral part of development. Glasier and Gülmezoglu even go as far as saying that "*the notion of reproductive health was born in Cairo in 1994*" [26].

That said, there was conflict over what was included as part of reproductive health. In fact, women's access to safe abortion was one of the prickliest issues at the ICPD, and negotiations over this topic proved particularly tricky. The conference organizers and attendees were wary of the issue, while religious organizations, states, and the media all attempted to promote their own agenda and frame [13].

Throughout the writing process, there was a struggle between pro-choice interest groups on the one hand and the Catholic church on the other. Though women's rights groups and NGOs lobbied throughout, the status of the Holy See as an observer state put it in a position of influence. This status also made the Roman Catholic Church unique among world religions [25]. That said, the Holy See was not alone in submitting reservations specifically on abortion. Many countries, including Honduras, Nicaragua, United Arab Emirates, Yemen, Dominican Republic, Guatemala, Malta, and Peru submitted oral and written reservations on the PoA statement on abortion [27].

As a result of these conflicts, the ICPD only went as far as to recognize unsafe abortions as a major public health concern [28]. ICPD framers chose to address reproductive rights through ambiguous language that at times was even inconsistent. It simultaneously phrased reproductive rights as a critical part of development, while denying abortion as an inherent part of reproductive health. Given that yearly millions of women around the world undergo abortions, both legal and illegal, denying the position of abortions as part of reproductive health ignores a reality of the day to day female experience.

Section 8, "Health, Morbidity, and Mortality", contains Paragraph 8.25, which is cited as summarizing and embodying the ICPD PoA statement on abortion:

> "*In no case should abortion be promoted as a method of family planning. All Governments and relevant intergovernmental and non-governmental organizations are urged to strengthen their commitment to women's health, to deal with the health impact of unsafe abortion as a major public health concern and to reduce the recourse to abortion through expanded and improved family-planning services. Prevention of unwanted pregnancies must always be given the highest priority and every attempt should be made to eliminate the need for abortion. Women who have unwanted pregnancies should have ready access to reliable information and compassionate counselling. Any measures or changes related to abortion within the health system can only be determined at the national or local level according to the national legislative process. In circumstances where abortion is not against the law, such abortion should be safe. In all cases, women should have access to quality services for the management of complications*

*arising from abortion. Post-abortion counselling, education and family-planning services should be offered promptly, which will also help to avoid repeat abortions*" [23].

The ICPD's position on abortion is considered one of its glaring weaknesses [29]. The language is indecisive, and the statements are inconsistent not only in terms of the minimum standards of abortion services each woman should enjoy but also in terms of its expectations from states and the extent to which it presents incentives for states to act in its own spirit concerning abortions. This vague wording and internal inconsistencies reflect the writers' efforts to simultaneously placate both those who called for expanded women's reproductive rights and autonomy and forces that opposed abortion.

Since 1994, the UN's World Health Organization has published a number of guidelines. In 2003 the WHO published the document *Safe abortion: technical and policy guidance for health systems*, followed by multiple revised additions including in 2012 and 2015. The next key policy pertaining to abortions emanating from the UN, though, was the Sustainable Development Goals [30].

Adopted by world leaders in September 2015, the 17 Sustainable Development Goals (SDGs) of the 2030 Agenda for Sustainable Development came into force in January 2016. This strategy of avoiding conclusive language in addressing reproductive rights seems to have been adopted in formulating the SDGs as well. Reproductive rights are addressed in SDG 3—"Good health and wellbeing"—in three different targets. Target 3.1 is to "*by 2030, reduce the global maternal mortality ratio to less than 70 per 100,000 live births*". While this doesn't name reproductive health, reproductive health is an integral part of maternal health and healthy pregnancies and births.

Target 3.7 addresses reproductive health most directly, aiming to "*by 2030, ensure universal access to sexual and reproductive health-care services, including for family planning, information and education, and the integration of reproductive health into national strategies and programme*s". Indicator 3.7.1 for this target is the "*proportion of women of reproductive age (aged 15–49 years) who have their need for family planning satisfied with modern methods*". Finally, target 3.8 is to "*achieve universal health coverage, including financial risk protection, access to quality essential health-care services and access to safe, effective, quality and affordable essential medicines and vaccines for all*". Its indicator, 3.8.1, specifies reproductive and maternal health coverage [30].

These targets and indicators at no point define what is part of reproductive health and what constitutes "reproductive rights", "reproductive healthcare", and "access" to reproductive services. The term "modern methods" of family planning, featured in indicator 3.7.1, is especially peculiar given the UN precedent to exclude abortion as a part of reproductive health and family planning. Replicating the strategy used in the ICPD PoA, the SDG framers attempted to simultaneously promote the idea of reproductive health and rights, without taking an alienating stance on any of its parts. The SDGs give no concrete guidelines for how to achieve these goals [30]. This desire to appease all parties prevents UN officials from giving clear instructions that might actually facilitate their objectives.

## 4. The Maputo Protocol: Modeling IGO Decisiveness

How can intergovernmental organizations make decisive guidelines and protocols within their diverse forums? If IGOs put such an emphasis on consensus, how can they balance the disparate interests of their members in creating effective guidelines for women's health? To our knowledge, one organization so far has risen to the challenge. In 2003, the AU published the Protocol to the African Charter on Human and Peoples' Rights on the Rights of Women in Africa, also known as the Maputo Protocol.

The Maputo Protocol is a legally binding supplement to the African Charter on Human and People's Rights of the African Union. It was adopted in Maputo, Mozambique, in July 2003 and entered into effect in November 2005 [31]. The document aimed to correct weaknesses in the African Charter on Human and People's Rights regarding women's rights [32]. The Protocol requires not just signing but also ratification. For states that ratify the Protocol, it is a legally binding treaty.

As of November 2019, 49 of the 55 AU countries have signed the Protocol. 42 countries have ratified. The only countries that have yet to sign are Botswana, Cape Verde, Egypt, Malawi, Morocco,

and Mauritania. Sao Tome and Principe ratified in April 2019, making it the most recent country to ratify. Ethiopia and Tunisia both ratified in 2018, though Ethiopia signed in 2004, and Tunisia in 2015 [33,34].

The AU Protocol is considered one of the world's most progressive treaties on women's rights. It addresses specific forms of women's rights' violations found in parts of Africa, in addition to more global gender equality issues [35]. Women's issues addressed include discrimination against women, elimination of harmful practices, marriage and divorce, reproductive rights, widows' rights, and rights of inheritance inter alia.

The Maputo Protocol is noted for its groundbreaking approach to reproductive rights. It applies a reproductive and public health paradigm to African abortion law, rather than the traditional crime and punishment model. For the drafters of the Maputo Protocol, unsafe abortion was a clear public health issue [36]. Africa has the highest rate of unsafe abortions and abortion-related deaths, accounting for 62% of unsafe abortion-related deaths worldwide. While in developing regions an estimated 30 women die for every 100,000 unsafe abortions, sub-Saharan Africa's rate of death is 17 times higher, with 520 women dying for every 100,000 unsafe abortions [37]. In the framing of the Maputo Protocol, drafters were sensitive to abortion as a public health concern.

The Protocol went much further than previous IGO efforts to define and promote reproductive rights, specifying a number of conditions under which abortion is a woman's human right [29,38]. Article 14.2, Section C states that:

*" . . . state parties should take all appropriate measures to . . . protect the reproductive rights of women by authorising medical abortion in cases of sexual assault, rape, incest, and where the continued pregnancy endangers the mental and physical health of the mother or the life of the mother or the foetus"* [33].

While it doesn't require states to legalize abortion on demand, it lays out a number of scenarios under which abortion must be legalized. This is the first time in international human rights law that a right to abortion has been enshrined in legal code. It is also the first time abortion is specified as a human right and as an enforceable obligation [38–40].

There are a few reasons we could guess that the AU drafters had more success addressing reproductive rights. First, the drafting process was longer and gave more room for civil society groups to review drafts, give input, and lobby. This created a less pressured atmosphere conceivably made it more likely to reach compromise [13].

Second, compared to the drafting process in the UN, the AU drafters were more willing to embrace conflict. The piecemeal fashion in which the Protocol has been signed and ratified over the last 17 years demonstrates that it was never created in order to achieve universal consensus. Instead, the goal was to determine a specific agenda and standard within the AU. Had consensus been a top priority, the drafters might have avoided any controversial or groundbreaking stance that would be a barrier to acceptance. The Protocol has correlated with increased rates of change in permissiveness of African abortion policy over time [13].

Finally, the writers of the Maputo Protocol might have been more successful in avoiding the politicization of reproductive rights because there were other women's rights issues that were considered much more controversial. For example, opposition to abortion rights was minor in comparison to the resistance generated against the effort to ban polygamy [39]. In general, African women's rights activists have found issues such as polygamy and child marriage much more difficult to address and change than issues of reproductive rights [38,41].

The precise reason notwithstanding, it is clear that within IGO forums, it is possible to create decisive statements on reproductive rights that both define the elements of reproductive health and women's reproductive rights. It might not be possible to do so without controversy, and it might be a barrier to universal acceptance. That said, clear guidelines and an unambiguous stance have been found to be more effective in making change on the ground in the health policies countries decide to adopt [42].

It is interesting to note that even in this progressive document, issues of different types of needs as a result of intersectionality are ignored. For example, article 23 of the Maputo Protocol addresses the "special protection of women with disabilities" [30]. While it mentions their right to freedom from sexual abuse and right to be treated with dignity, there is no mention of their right to healthcare or reproductive services.

## 5. Discussion

The strategy that the UN adopts in order to avoid politicization of reproductive rights seemingly undermines its impact on global public health. Ultimately, the SDGs' ambiguity could hurt the ability of the Goals to give clear guidelines to countries for how to define, provide, and promote public health. This leads to two main questions for the framers of the SDGs and the framers of future similar UN goals. First, how is it undermining goal attainment? Second, what alternative strategies can help overcome this challenge?

We propose three key reasons that reproductive health needs to be depoliticized for future UN efforts to remedy the shortcomings of SDG3. First, UN protocols create precedents for local grassroots organizations to promote gender equality and women's health rights. They can serve norm and policy dissemination by strengthening local NGOs and civil society organizations [43]. The precedents created by IGO research bodies, position papers, projects, and agreements [44] all provide tools for local NGOs and citizens to demand change at a country level [45,46]. Second, UN protocols can legitimize policy positions and create new norm precedents that can trickle down into local politics. Local players can use IGO protocols as grounds for policy change [47,48], as a source of information or as a blueprint for national policy.

Third, the politicization of reproductive health doesn't just limit the impact of SDG3 but also impacts other SDGs and therefore the much wider SDG agenda. SDG5 is "gender equality" and the empowerment of women and goals. Target 5.6 aims to:

"*Ensure universal access to sexual and reproductive health and reproductive rights as agreed in accordance with the Programme of Action of the International Conference on Population and Development and the Beijing Platform for Action and the outcome documents of their review conferences*" [30].

The indicators for this target are "*proportion of women aged 15–49 years who make their own informed decisions regarding sexual relations, contraceptive use and reproductive health care*" and "*Number of countries with laws and regulations that guarantee women aged 15–49 years access to sexual and reproductive health care, information and education*".

Just like in SDG3, these targets and indicators in SDG5 leave several open questions. What should be a part of universal access to reproductive health? Is abortion part of contraception and reproductive healthcare? Is abortion an inherent part of access to sexual and reproductive healthcare, and therefore should it be protected as a woman's right, as seen in the Maputo Protocol? The same ambiguities that plague the area of reproductive health in SDG3 reappear in SDG5.

This leads to the second question; how can UN framers overcome the politicization of women's reproductive health? There are two alternative strategies that could be adopted. First, UN drafters might return to the different frames often found when addressing reproductive rights in media. The politicization of reproductive rights is the outcome of it being in the overlap between morality policy and feminist policy. Debates in UN forums have been between actors who adopt these two disparate frames. The third frame though, the medical frame, has been found to be more effective both at the local and at the IGO level. Therefore, UN organizations must more fully commit to seeing reproductive health as a public health and medical issue. If they allow the other two frames to dominate the discussion, they are less likely to achieve the stated SDG3 goal of good health and wellbeing. A medical frame might make it easier to address issues of access to reproductive health and definitions of reproductive rights for different groups that arise from intersectional identities. Though the medical

frame removes the debate on reproductive rights from the human rights frame often used to discuss minority group rights, it can facilitate a perspective that looks at the medical needs of different groups and looks at how different groups access health services.

Another strategy for addressing reproductive rights might be to embrace a certain level of conflict as an unavoidable part of engendering a more meaningful change. As part of adopting a medical frame, UN framers might have to be more open to conflict and to creating clear definitions of what is a "reproductive right", what is "access", and what are considered inherent parts of reproductive health. This might mean including procedures that conflict with certain religious or cultural groups but are inherently a part of women's family planning and reproductive health strategies worldwide. Framers might have to forgo consensus in order to render results in their targets. Countries that oppose procedures such as abortion will ignore UN suggestions anyway. For countries that are on the fence, however, specifying clear guidelines and rights in UN protocols could make all the difference at the local level.

## 6. Conclusions

The politicization of women's reproductive health, as a result of its overlap as both a morality and feminist policy issue, creates unique hurdles to achieving UN SDG3. This article explored how reproductive health's overlap between the two theoretical categories has impacted UN work. In an effort to please all and maintain consensus, the UN has avoided taking stances on women's reproductive health and has left countries without the clear guidelines necessary to achieve health goals.

This is not a new problem in UN work. It was seen close to 30 years ago, when Catholic and women's groups clashed during the 1994 ICPD conference. ICPD PoA framers overcame the conflict by adopting a strategy of ambiguous language. This strategy was adopted yet again when framing the SDGs.

Several alternative strategies are also available. The UN can look to the AU Maputo Protocol for precedent and to see how another major IGO succeeded in creating specific guidelines for reproductive health. The Maputo Protocol, written only eight years after the ICPD PoA, was groundbreaking in laying out detailed conditions for reproductive health and abortion rights. AU framers did this by both allowing for the input of civil society groups as well as applying a strategy that accepted inevitable conflict and difference of opinion. Such a strategy could be key to achieving UN SDGs, as well as similar future programs. Countries and civil society groups rely on UN goals to create precedents and guidelines for local policy. Without understandable recommendations, local players are left to generate their own interpretations of SDG framer intentions.

Perhaps the best way to alter the debate in UN forums is to adopt a medical frame. This frame sees reproductive health as a public health issue and leaves it to the expertise of doctors and healthcare workers to define what is considered an inherent part of reproductive health. This of course will require UN bodies to accept a certain amount of conflict, as well as forgo universal consent. If they want to overcome the politicization of reproductive health and make greater progress towards achieving both SDG3 and SDG5, this might be the most viable strategy.

**Author Contributions:** U.S. and A.F.-R. conceived and designed the theoretical framework, analyzed the data, and wrote the paper. Both authors have read and agreed to the published version of the manuscript.

**Funding:** His research received no external funding.

**Conflicts of Interest:** The authors declare no conflict of interest.

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
