# Peer review of "The Politicization of Women’s Health and Wellbeing"

_sustainability, doi:10.3390/su12093593_

Round 1

Reviewer 1 Report

In my opinion this is a fine manuscript that deserves publication when three problems are resolved:

  1. Authors claim “We will then trace how the conflict between those categories [morality policy and feminist policy] has played out in the history of UN work including in SDG3” (page 2). In the relevant section “The clash between morality and feminist policy advocates in UN work and SDGs” authors first focus on the ICPD in 1994 and then jump directly to the SDGs at the top of page 5, which were adopted 21 years later. Have there been no relevant discussions or decisions in UN bodies in the meantime? If so, the exclusive focus on ICPD and SDG is correct, but it should be justified. In addition, the year the SDGs were set, should be mentioned.
  2. On page two the authors write “Based on lessons learned from the success of the AU, UN bodies might adopt two steps; first they can fully commit to a medical frame when addressing reproductive rights. Second, they can replace a strategy that aims for universal consensus with one that is more open to conflict”. In the relevant section “The Maputo Protocol: Modeling IGO decisiveness” they mention the willingness to embrace conflict as one reason for the Maputo Protocol's clear position on reproductive rights. They also mention a longer drafting process that gave civil society more room to influence and the fact that other women's rights (a ban on polygamy and child marriage) were more controversial than reproductive rights. But I cannot find a reference to the medical frame. The authors should make this reference clearer in the revision of their manuscript.
  3. On the one hand, the authors postulate a greater willingness to engage in conflict as a strategy for the success of future UN negotiations on reproductive rights; on the other hand, they state "that reproductive health needs to be depoliticized for UN efforts to remedy the shortcomings of SDG3" (page 6). In my view, this does not fit together, since politicization and not depoliticization of an issue is often accompanied by more conflicts. The authors should make it clearer why willingness to embrace conflict and depoliticization should be compatible.

Finally, I have a smaller point, which strictly speaking is not part of the focus of the manuscript. What were the consequences of the Maputo Protocol? Did the signatory states subsequently liberalize their abortion laws or improve practical access to abortion, or has it remained a mere declaration of intent?

Author Response

R1

  1. Authors claim “We will then trace how the conflict between those categories [morality policy and feminist policy] has played out in the history of UN work including in SDG3” (page 2). In the relevant section “The clash between morality and feminist policy advocates in UN work and SDGs” authors first focus on the ICPD in 1994 and then jump directly to the SDGs at the top of page 5, which were adopted 21 years later. Have there been no relevant discussions or decisions in UN bodies in the meantime? If so, the exclusive focus on ICPD and SDG is correct, but it should be justified. In addition, the year the SDGs were set, should be mentioned.

To address these issues, on p. 12 we add:

The World Health Organization published in the following years several sets of guidelines for abortions. In 2012, for instance, the WHO published Safe abortion: technical and policy guidance for health systems. The next key ruling pertaining to abortions emanating from the UN was the Sustainable Development Goals.

Adopted by world leaders in September 2015, the 17 Sustainable Development Goals (SDGs) of the 2030 Agenda for Sustainable Development came into force in January 2016.

  1. On page two the authors write “Based on lessons learned from the success of the AU, UN bodies might adopt two steps; first they can fully commit to a medical frame when addressing reproductive rights. Second, they can replace a strategy that aims for universal consensus with one that is more open to conflict”. In the relevant section “The Maputo Protocol: Modeling IGO decisiveness” they mention the willingness to embrace conflict as one reason for the Maputo Protocol's clear position on reproductive rights. They also mention a longer drafting process that gave civil society more room to influence and the fact that other women's rights (a ban on polygamy and child marriage) were more controversial than reproductive rights. But I cannot find a reference to the medical frame. The authors should make this reference clearer in the revision of their manuscript.

On p. 12-13, we highlight the relation between the Protocol and the extremely high rate of unsafe abortions in Africa, which played a role in the drafting. Specifically, we write:

Part of the motivation for the drafting of the Protocol was in response to Africa having, by-far, the highest global rate of unsafe abortions and abortion related injuries and deaths. In that sense, the medical frame played a role. Indeed, Hildebrandt (2015) provides several important illustrations for the medical framework, in particular in places like African countries where unsafe abortions are so prevalent. Rather than doctors, poor women, he asserts “rely on coat hangers”, and may suffer prosecution if hospitalized due to injuries from clandestine abortions. Moreover, fearing criminal prosecution, health-care providers may opt to deny certain types of medical treatments (Htun 2003).

  1. On the one hand, the authors postulate a greater willingness to engage in conflict as a strategy for the success of future UN negotiations on reproductive rights; on the other hand, they state "that reproductive health needs to be depoliticized for UN efforts to remedy the shortcomings of SDG3" (page 6). In my view, this does not fit together, since politicization and not depoliticization of an issue is often accompanied by more conflicts. The authors should make it clearer why willingness to embrace conflict and depoliticization should be compatible.

We clarified this point in the discussion (p.17). This is correct that depoliticization should inherently mean less conflict, and therefore embracing conflict would imply continued politicization. Therefore, we clarified that these are two separate strategies- 1) depoliticization through the medical frame 2) accepting conflict as an inherent part of the process.

  1. Finally, I have a smaller point, which strictly speaking is not part of the focus of the manuscript. What were the consequences of the Maputo Protocol? Did the signatory states subsequently liberalize their abortion laws or improve practical access to abortion, or has it remained a mere declaration of intent?

The answer is that there is evidence for change in African countries (though we cannot establish causality). We deal with this issue elsewhere, which we now indicate on p. 14 of the revised manuscript:

The Protocol has correlated with increased rate of change in permissiveness of African abortion policy over time (Sommer and Forman-Rabinovici, 2019).

Reviewer 2 Report

Dear Authors,
Thank you for the opportunity to review your manuscript. It's an important topic and your study intends to address the challenge of women's health and SDGs. My comments and suggestions below are meant to point to some ways that your submission can be strengthened so that others can benefit from your analysis. 

I understand that your primary focus is on SDG 3. I am also certain you understand how the SDGs are interconnected and how SDG3 is closely linked to SDG5, which you briefly touched on in your paper. Coming from a disability rights background, I am particularly concerned that your analysis does not address the health and well-being concern of women and girls with disabilities. As you weave the 2003 Maputo Protocol into your discussion, I suggest you consider its Article 23 that provides for “Special Protection of Women with Disabilities” in your analysis. Given that women with disabilities endure double discrimination, both for being women and for living with a disability, and given that women with disabilities face particular difficulties in gaining access to health care, I strongly believe your analysis and conclusions will be enriched by that perspective. Both the 2006 UN Convention on the Rights of Persons with Disabilities (UN CRPD) and disability rights scholarship have strongly advocated against the use of the medical frame. I am not sure how you can reconcile including women and girls with disabilities into your analysis with maintaining your suggestion of the medical frame for women's reproductive health, but I am certain it will make your discussion more inclusive and robust.

- references and citations are not in accordance with the MDPI's guidelines.

- minor grammatical and syntax errors; maintain consistency in using the serial comma.

Author Response

R2

I understand that your primary focus is on SDG 3. I am also certain you understand how the SDGs are interconnected and how SDG3 is closely linked to SDG5, which you briefly touched on in your paper. Coming from a disability rights background, I am particularly concerned that your analysis does not address the health and well-being concern of women and girls with disabilities. As you weave the 2003 Maputo Protocol into your discussion, I suggest you consider its Article 23 that provides for “Special Protection of Women with Disabilities” in your analysis. Given that women with disabilities endure double discrimination, both for being women and for living with a disability, and given that women with disabilities face particular difficulties in gaining access to health care, I strongly believe your analysis and conclusions will be enriched by that perspective. Both the 2006 UN Convention on the Rights of Persons with Disabilities (UN CRPD) and disability rights scholarship have strongly advocated against the use of the medical frame. I am not sure how you can reconcile including women and girls with disabilities into your analysis with maintaining your suggestion of the medical frame for women's reproductive health, but I am certain it will make your discussion more inclusive and robust.

This is an important point, which we revised the manuscript to recognize. Please see on p. 18 of the revised manuscript:

We say that, while keeping in mind that the medical frame is not without its weaker sides. Both the 2006 UN Convention on the Rights of Persons with Disabilities (UN CRPD) as well as some scholarship have advocated against the use of the medical frame, which may put women with disabilities at double the disadvantage (see also Art. 23 of the Maputo Protocol, “Special Protection of Women with Disabilities”). While we recognize that those issues would have to be reconciled, the scope of this manuscript would not allow to fully develop this point, which will be elaborated in future work.

- references and citations are not in accordance with the MDPI's guidelines

We believe references were already updated. Please let us know if there’s any additional changes necessary.

- minor grammatical and syntax errors; maintain consistency in using the serial comma

We read through the article, and made changes where we believe necessary. Please let us know if there’s any additional changes necessary.

Round 2

Reviewer 2 Report

Dear Authors,

Thank you for revising the manuscript to the best of your ability. I understand that a detailed discussion of the reproductive issues of disabled women is outside the scope of your paper and may be saved for later. But at least a brief discussion of such issues specific to women with disabilities is needed for a well-rounded discussion of the framing of women's reproductive rights and SDGs. One suggestion for it may be the section of the Maputo Protocol, and more specifically its Art.23, or elsewhere in the paper as you see fit. Deliberately dismissing women with disabilities from your analysis because such a discussion may weaken your argument is not a good scholarship practice. It is akin to excluding a dataset that does not support your hypothesis. I saw your mention of the UNCRPD at the very end of the manuscript. It makes the Conclusions section confusing, as in the present form, it introduces a new idea (women with disabilities and UNCRPD) that does not follow from the paper's previous analysis. My suggestion would be to extract such essential and relevant information from the UNCRPD and The Maputo Protocol and include these insights throughout your paper. 

A quick point on the Citations and References: the newly submitted version of the manuscript still does not conform to the Instructions for Authors (i.e., numbered, placed in square brackets, headings/subheadings, according to the template). 

Author Response

R2

  1.  I understand that a detailed discussion of the reproductive issues of disabled women is outside the scope of your paper and may be saved for later. But at least a brief discussion of such issues specific to women with disabilities is needed for a well-rounded discussion of the framing of women's reproductive rights and SDGs. One suggestion for it may be the section of the Maputo Protocol, and more specifically its Art. 23, or elsewhere in the paper as you see fit. Deliberately dismissing women with disabilities from your analysis because such a discussion may weaken your argument is not a good scholarship practice. It is akin to excluding a dataset that does not support your hypothesis. I saw your mention of the UNCRPD at the very end of the manuscript. It makes the Conclusions section confusing, as in the present form, it introduces a new idea (women with disabilities and UNCRPD) that does not follow from the paper's previous analysis. My suggestion would be to extract such essential and relevant information from the UNCRPD and The Maputo Protocol and include these insights throughout your paper. 

We have taken this topic and used it as a springboard to address and exemplify how the clash between feminist and morality policy has the additional problem of blinding policy makers to issues of intersectionality. Indeed, neither of these frames accounts for the double stigma that women with disabilities experience when it comes to reproductive rights, nor does it allow policy makers consider how different groups access healthcare.

We have incorporated this into our discussion at a number of points. First, in the section titled "Reproductive health and the clash between morality policy and feminist policy" we write that:

Both approaches also often fail to account for questions of intersectionality, such as how motherhood, pregnancy and abortion are perceived by and for different groups.  For example, disabled women experience a double-stigma, in which as women they are expected to raise children, but as people with disabilities they are often viewed as a-sexual, or incapable of fulfilling traditional feminine roles [46]. The feminist movement has at times failed to account for the needs of women with disabilities or the rights of the disabled [47], while morality-based movements fail to recognize unique forms of reproductive healthcare they might require. Additionally, neither movement accounts for the difficulty of access to healthcare that many disabled women experience [48,49].

In the section on the Maputo Protocol we return to this issue, showing how even in this protocol, there is a disconnect between understandings of women's reproductive rights and the unique standing of disabled women. We write:

            It is interesting to note that even in this progressive document, issues of different types of needs as a result of intersectionality are ignored. For example, article 23 of the Maputo Protocol addresses the "special protection of women with disabilities" [30]. While it mentions their right to freedom from sexual abuse and right to be treated with dignity, there is no mention of their right to healthcare or reproductive services.

In the discussion though, we look at this issue from the perspective of the medical frame. The medical frame can actually address this issue as well. We write:

A medical frame might make it easier to address issues of access to reproductive health, and definitions of reproductive rights for different groups that arise from intersectional identities. Though the medical frame removes the debate on reproductive rights from the human rights frame often used to discuss minority group rights, it can facilitate a perspective that looks at the medical needs of different groups and looks at how different groups access health services. 

We removed discussion of the UNCRPD from the Conclusions.

References and citations are now in accordance with the MDPI's guidelines (numbered, placed in square brackets, according to the template)

Round 3

Reviewer 2 Report

Dear Authors,

Thanks for your persistence and resubmission. I admire and applaud you for taking the previous comments into consideration and working them into the manuscript. I could see how your argument became more robust and more inclusive. I hope you continue including the population with disabilities in your analysis in the future, when appropriate.

Good wishes to you in your work and pursuits.